REGISTERED REPORT PROTOCOL

# Registered report protocol: Stress testing predictive models of ideological prejudice

**Jordan L. Thompson**[1]*, **Abigail L. Cassario**[2], **Shree Vallabha**[2], **Samantha A. Gnall**[1], **Sada Rice**[1], **Prachi Solanki**[2], **Alejandro Carrillo**[2], **Mark J. Brandt**[2], **Geoffrey A. Wetherell**[1]

1 Department of Psychology, Florida Atlantic University, Boca Raton, Florida, United States of America,
2 Department of Psychology, Michigan State University, East Lansing, Michigan, United States of America

* jordanthomps2021@fau.edu

This is a Registered Report and may have an associated publication; please check the article page on the journal site for any related articles.

**Data Availability Statement:** The data used to write the registered report protocol and create the

## Abstract

In this registered report, we propose to stress-test existing models for predicting the ideology-prejudice association, which varies in size and direction across target groups. Previous models of this relationship use the perceived ideology, status, and choice in group membership of target groups to predict the ideology-prejudice association across target groups. These analyses show that models using only the perceived ideology of the target group are more accurate and parsimonious in predicting the ideology-prejudice association than models using perceived status, choice, and all of the characteristics in a single model. Here, we stress-test the original models by testing the models' predictive utility with new measures of explicit prejudice, a comparative operationalization of prejudice, the Implicit Association Test, and additional target groups. In Study 1, we propose to directly test the previous models using an absolute measure of prejudice that closely resembles the measure used in the original study. This will tell us if the models replicate with distinct, yet conceptually similar measures of prejudice. In Study 2, we propose to develop new ideology-prejudice models for a comparative operationalization of prejudice using both explicit measures and the Implicit Association Test. We will then test these new models using data from the Ideology 2.0 project collected by Project Implicit. We do not have full access to this data yet, but upon acceptance of our Stage 1 registered report, we will gain access to the complete dataset. Currently, we have access to an exploratory subset of the data that we use to demonstrate the feasibility of the study, but its limited number of target groups prevents conclusions from being made.

## Introduction

How can we predict whether liberals or conservatives will express more prejudice toward a group? Brandt sought to predict the ideology-prejudice association for multiple target groups with models using ideology, perceived group status, and perceived choice of group membership as predictors [1]. The goal was to create models that make accurate predictions of the ideology-prejudice association toward a variety of target groups and to help us understand which theoretical perspectives are most useful when predicting ideological prejudice. We replicate and extend this work here.

tables and figures in this paper on our OSF page. The data can be found in the "Data" folder. Here is the DOI: https://doi.org/10.17605/OSF.IO/BUWP7 The proposed registered report proposes collecting data about people's perceptions of social groups' status, ideology, and choice. This anonymized novel data collection effort is publicly shared in its entirety on the OSF page. We are also proposing the use of the Ideology 2.0 dataset that was collected and is maintained by Project Implicit. We only have access to this data upon acceptance of the Stage 1 version of this registered report. Because we are not the owners of the data, we are not able to share the data immediately. The Ideology 2.0 team has placed this restriction on the data so that it can be used for registered reports (if it were posted publicly, then scholars would already have access to it, decreasing its potential for a registered report). Therefore, we cannot share this data immediately. However, we will add this data to our data sharing repository on the OSF as soon as the Ideology 2.0 team allows for this. As part of our study, we essentially create a new database on ideology-prejudice associations across groups and measures. We use these to compare with model predictions. The database of these associations are shared on the OSF page. We will also share all materials and code for the entire project on the OSF page.

**Funding:** The authors received no specific funding for this work.

**Competing interests:** The authors have declared that no competing interests exist.

In effort to forecast prejudice toward various groups, the prejudice literature has offered three main predictors: the ideological position of a group, the group's status, and the extent to which membership in this group is a choice. Research indicates that people dislike groups that hold ideological positions dissimilar from their own [2–4], indicating that perceived ideological dissimilarity should be associated with ideological prejudice. For example, Chambers and colleagues found that conservatives prefer traditionally conservative groups, and liberals prefer liberal groups [3]. Other works suggest that conservatives may be more prejudiced in general [5], or that conservatives may be more prejudiced against low-status groups in particular [6]. For example, research suggests that people may be prejudiced toward privileged groups as well as marginalized groups [6], and political conservatism has been shown to relate to prejudice against a variety of low-status targets [7]. Another perspective proposes that perceived choice regarding group membership is valued by conservatives as it helps to define boundaries between groups [7, 8]. Additional work indicates that people with lower levels of cognitive ability tend to be more prejudiced toward groups with little choice in group membership [9]. Brandt [1] compared these three perspectives to one another. In this project, we build on and stress-test the work of Brandt [1] to try and predict the size and direction of the ideology-prejudice association.

The prejudice and ideology literatures do not offer many concrete methodological suggestions for predicting the magnitude and direction of the effect of ideology on prejudice against specific groups. As an example, widely cited dual process models of prejudice suggest that a desire for social conformity and belief in a dangerous world predict right-wing authoritarian (RWA) attitudes, whereas the belief the world is a competitive jungle predicts social dominance, both of which predict prejudice [10]. Despite providing a theoretical rationale for the relationship between manifestations of ideology (i.e., RWA) and prejudice towards certain groups, this perspective does not provide scaffolding to predict the specific level of prejudice a person will exhibit towards a variety of target groups, in addition to examining a somewhat limited set of targets. This is a gap that should be filled because there is both practical and theoretical value in models that can predict the ideology-prejudice association across targets [1].

First, a predictive model can help scholars examine whether the association between ideology and prejudice will be stronger and in what direction depending on different characteristics of a target group (e.g., status, perceived ideology). Different combinations of characteristics may yield different predictions of the strength and direction of the ideology-prejudice association, which allows for greater model predictive accuracy.

Second, these models can be used to make predictions using new samples and target groups. This underscores the generalizability of such models, allowing scholars to anticipate associations before data is collected. Ideology-prejudice association predictions provide estimates of effect size, which means that they can be used to conduct power analyses prior to collecting data and can serve as a theoretical starting point when proposing new models. Likewise, predictive models have theoretical implications because predictions of effect size can be useful in falsifying hypotheses and comparing rival models. Indeed, recent research in psychology has used predictive models to predict diverse outcomes, such as changes in trust [11], romantic interest [12], and success in psychotherapy [13]. Here we further develop predictive models of the ideology-prejudice association.

## Predicting ideological prejudice

Brandt [1] built models of the ideology-prejudice association using data from the 2012 wave of the American National Election Studies (ANES) and then tested the predictive accuracy of those models in new samples with new target groups. The ideology-prejudice association was

**Table 1. Predictive equations of prejudice from Brandt's [1] work and preliminary MSEs (Study 1).**

| Model Name | Theoretical Implication | Model | MSE (SD) Estimate Across Outcomes | MSE (SD) for Actual Prejudice Outcome | MSE (SD) for Gut Prejudice Outcome | MSE (SD) for Positive Prejudice Outcome | MSE (SD) for Negative Prejudice Outcome |
|---|---|---|---|---|---|---|---|
| ideology-only | Ideological differences explain ideology-prejudice association | $\hat{y} = 0.022 - 1.420$ (ideology) | .01 (.02) | .02 (.01) | .02 (.02) | .04 (.05) | .03 (.03) |
| status-only | Status differences explain ideology-prejudice association | $\hat{y} = 0.001 - 0.846$ (status) | .14 (.15) | .20 (.21) | .22 (.23) | .08 (.10) | .10 (.10) |
| choice-only | Choice differences explain ideology-prejudice association | $\hat{y} = 0.041 - 0.398$ (choice) | .16 (.16) | .22 (.22) | .24 (.24) | .10 (.11) | .09 (.11) |
| ideology, status, and choice | A combination of group characteristics explains ideology-prejudice association | $\hat{y} = 0.016 - 1.505$ (ideology) $+ 0.128$ (status) $+ 0.072$ (choice) | .03 (.04) | .03 (.02) | .04 (.04) | .06 (.07) | .03 (.03) |
| null | Group characteristics do not explain ideology-prejudice association | $\hat{y} = 0$ | .16 (.16) | .22 (.22) | .24 (.24) | .08 (.09) | .11 (.13) |

Table 1 includes Brandt's original equations [1]. The MSEs and SDs for each model and for each type of measure are also presented in the table.

estimated for 24 groups in the 2012 ANES, including political groups, religious groups, socio-economic groups, and racial/ethnic groups. Each target group was also rated by a separate sample in terms of their perceived ideology, status, and choice in group membership. These group characteristics were then used to build models of the ideology-prejudice association. For example, an ideology-only model used only the perceived ideology of the target group to predict the size and direction of the ideology-prejudice association. Other models included other group characteristics (e.g., a status-only model) or combinations of characteristics (e.g., ideology, status, and choice). The estimates from these models were then used to predict the size and direction of the ideology-prejudice association in a variety of additional samples and target groups.

Brandt compared the predicted association with the observed association between ideology and prejudice using Mean Squared Errors (MSE) [1]. Specifically, to generate predicted values, he used data from the 2012 ANES to estimate the ideology-prejudice association for 24 groups. Then, he built models using measures of each groups' perceived ideology, status, and choice (obtained from separate samples). The estimates from these models were used as the *predicted* ideology-prejudice association. The *observed* ideology-prejudice associations were obtained by estimating the ideology-prejudice association for each target group in each of four studies. Then, Brandt compared the predicted ideology-prejudice associations with the observed ideology-prejudice associations (residuals), squared them, and then found their average for each model. The model with the lowest MSE was the best-fitting model (see Table 1 for the four models we focus on, plus a null model).

Brandt's [1] predictive analyses revealed that models that included only ideology or that included ideology, status, and choice group characteristics were the most accurate in predicting the ideology-prejudice association in new data. These models were more accurate than models that only included status or only included choice (for a replication, see [14]). Additional models included perceived conventionalism. Because perceived conventionalism and ideology were highly overlapping, Brandt [1] focused on the ideology models. We do the same

here. This pattern of results suggests that the ideology-only model, the most accurate and most parsimonious model, was the best model because it yielded the lowest MSE values. Although perceptions of group status and choice have often been related to prejudice [5, 8, 15–17], the perceived ideology of the target group may be the biggest factor for understanding towards which groups liberals and conservatives express the most prejudice toward.

## Stress-testing existing models

The purpose of this project is to stress-test Brandt's [1] original models using alternative measures of prejudice, including alternative explicit measures (i.e., relative measures of explicit prejudice), and reaction time measures (i.e., Implicit Association Test scores [18]). This addresses a key shortcoming of the original study. In particular, the original study used only feeling thermometers as an explicit measure of prejudice. Although feeling thermometers are common measures of prejudice (e.g., [19–21]), they are just one possible measure of group-based attitudes. The predictive accuracy of the models might be limited to feeling thermometers. If so, this would limit the utility of the predictive models to only studies that use feeling thermometers. This is a potentially substantial limitation given the diversity of prejudice measures that exist. However, if the predictive models work well with alternative measures of prejudice, i.e., the explicit and reaction time measures, it suggests that the model is much more generalizable. This would mean that this model could be used for different types of prejudice measures.

In addition, the original models were built and tested using an absolute definition and operationalization of prejudice. That is, prejudice was defined as a negative group-based attitude (see also [22, 23]). This means that a person who expresses negative attitudes about both men and women would be considered prejudiced against both men and women. Although this is a widely used definition and operationalization of prejudice, it is not the only definition. Another widely used definition and operationalization of prejudice is a relative definition. That is, prejudice is defined as a negative attitude about a group *relative* to one's attitudes about another group (see also [6, 18, 24]). This definition captures notions of bias and differential attitudes that are often associated with the concept of prejudice. Consider a person who expresses negative attitudes about both men and women. Under the relative definition of prejudice, they would not necessarily be considered prejudiced because there is no difference in their attitudes about men and women. In contrast, a person who expresses more negative attitudes about women relative to their attitudes about men would be considered prejudiced toward women. One contribution of the current work is that we test both operationalizations of prejudice.

The traditional IAT, for example, is a relative comparison between two groups. It assesses differences in participants' reaction times when associating groups with positive and negative words [18]. Similarly, explicit measures that ask respondents to choose which group they like best or look at the difference in negative feelings against one group compared to another would also be relative measures of prejudice. Brandt's [1] models were not made for such relative measures. Therefore, in Study 2, when we test relative measures, we will first build new models (conceptually based on those in Table 1) for relative measures of prejudice. After building models for relative prejudice, we will test the models on data the models have not previously seen.

## The current studies

Here we propose testing four key models from Brandt [1], in addition to a null model (see Table 1) using a large dataset. Schmidt and colleagues [25] have issued a call for registered

reports and have agreed to provide us with a new and very large (N = 261,119) dataset to conduct our analyses upon in-principle acceptance of a Stage 1 registered report. Studies 1 and 2 will use the Ideology 2.0 dataset [25].

We will replicate (Study 1) and extend (Study 2) Brandt's [1] work in two ways. First, in Study 1, we will use models derived from the original work to estimate the extent to which the ideology-absolute-prejudice relationships for each group in the Ideology 2.0 data are explained by the models. If the original models are robust, they should predict the ideology-absolute-prejudice relationship in the Ideology 2.0 dataset with different measures with relative accuracy. If they are not predictive of the ideology-prejudice association measured using alternative explicit measures of absolute prejudice, it will provide valuable insight into how ideology may relate to different types of prejudice and the boundary conditions of the models.

Study 2 goes beyond this. In this study, we will build new predictive models of *relative* prejudice. This contributes to the literature by adding conceptual and computational depth to existing work. We examine the predictive abilities of our models by leveraging the size of the Ideology 2.0 dataset to perform a train/test split [26]. The train/test method is a data analytic method that allows for model predictions by splitting a dataset into two parts. One part is the "training" set that is used to build the model and the other part is the "test" set that is used to test and validate the model. We provide a more extensive description of the train/test procedure in the Study 2 Method section. Following the general procedure of Brandt [1], we will first train new models using the perceived differences between target groups on perceptions of ideology, status, and choice to predict the association between participant ideology and relative prejudice toward the target groups (e.g., the difference in perceptions of Black and White people's perceived ideology predicting ideology's association with Black vs. White relative prejudice). After estimating these models, we will test their predictive accuracy in the test set of data. We will use these models to predict levels of relative prejudice toward each pair of target groups and compare the results of the models to the observed relationship between ideology and comparative prejudice in the data. This will allow us to generate new predictive models using a wider swath of prejudice measures.

For both studies, we need to obtain estimates of how people perceive the target groups. These are the basis for the models in Brandt [1] and are therefore necessary to estimate the models. Specifically, we need to know how the target groups are perceived in terms of their ideology, status, and choice. Some of the groups that we will include in our models were not included in prior work, so it is unknown how they are perceived on these characteristics. Therefore, we will collect new data to obtain these perceptions, which will be used for both Studies 1 and 2. Collecting this data is important because it allows us to examine whether the models derived from Brandt [1] generalize to new sets of target groups using different types of measures of prejudice.

## Method

We first describe the details of new data collection that we will undertake for assessing how various groups from the Ideology 2.0 dataset are perceived in terms of ideology, status, and choice. We will then describe the details of Study 1 and Study 2 that we undertake on the Ideology 2.0 dataset.

### Ethics approval

We have obtained ethics board approval to collect new data as well as use Schmidt's archival data [25] (BLINDED FOR REVIEW University Institutional Review Board protocol number 2044847–1).

Participants in the Ideology 2.0 study participated in an online study. They read a consent document and advanced to the screen with the study instructions only if they agreed to participate. Because of the nature of online studies, it was not possible to obtain written consent as any signatures obtained would inevitably be linked to specific participants. The researchers who own the dataset obtained approval from the University of Virginia's Institutional Review Board for this procedure.

In the new data collection, the consent document will precede the survey. Participants can read about the study and decide whether they would like to participate. If participants choose to participate, they can click an arrow button to advance to the survey (which will indicate informed consent). If someone chooses not to participate, they can close out of the survey with no penalty. The current protocol involves online data collection and does not contain any identifying or sensitive information. It is not possible for participants to provide a signature in this context and doing so would make participant data identifiable given that the signature would be linked to their survey responses. The Institutional Review Board at BLINDED FOR REVIEW University approved this procedure.

## I. New data collection

**Participants and procedure.**   We will administer the survey via Prolific. Participants can choose which studies they would like to complete on the Prolific website. They will read a short description of the study and click on the survey link if they are interested in participating. Potential participants will read a consent form before proceeding with the survey. Participants will fill out basic demographic information and then respond to the other survey items. The study should take about 10 minutes to complete.

We will recruit a sample of 100 people. We will recruit Prolific participants who have an approval rate of greater than 95 and are Americans who are living in America. We aim to recruit an equal number of men and women. Participants will be paid $2.00 for their participation.

**Measures.**   This new data collection will be used to obtain group ratings for multiple registered reports using the Ideology 2.0 data. For the purposes of this study, we will collect participants' perceptions of the ideological positions of the target groups, the status of the target groups, and the extent to which membership in the target group is a choice (we will use the measurements suggested by Brandt and Crawford [9] and used by Brandt [1]). Each participant will rate each target group on each characteristic.

We will include three groups in the new data collection (scientists, capitalists, and socialists) that will not be included in the main study because the main study does not include these groups (only the concepts, e.g., of "science"). Other members of our research team will also be using this data for a different project, so these three groups will be included in the new data collection study to conserve resources. There is no reason to think that including these additional groups will affect the group ratings. We will need 100 ratings for 21 target groups (18 groups from the main study + 3 additional) which means we need 2100 ratings in total. All survey items are contained in the S1 Appendix.

**Perceived ideology.**   Ideological positions of the target groups will be measured on a scale from 0 (*strongly liberal*) to 100 (*strongly conservative*). We will pair the target groups with the following statement from Chambers and colleagues [3], "For each group indicate whether you think the group is typically a liberal or conservative group."

**Perceptions of group status.**   Status of target groups will be measured on a scale from 0 (*low status*) to 100 (*high status*). Before asking about status of the target groups, we will present participants with this statement adapted from Fiske and colleagues [15] (modified by Brandt

and Crawford [9] and used by Brandt [1]): "Some groups in society have higher status. That is, they have more education, they have more prestigious jobs, and they are more economically successful than other groups. Some groups have lower status. That is, they have less education, less prestigious jobs, and are less economically successful than other groups. And, of course, some groups are more in the middle."

**Perceived choice.**   Choice will be measured on a scale from 0 (*not at all*) to 100 (*very much*). Participants will read this statement adapted from Haslam and Levy [27] (modified by Brandt and Crawford [9] and used by Brandt [1]): "Sometimes people have choice and control of whether they belong to a particular group. Other times, they do not have much choice and control over whether they belong to a particular group." Then, they will respond to the question, "To what extent can members of this group choose or control whether they actually belong to this group?" for each target group in the study.

**Demographics.**   Additionally, we will include demographic measures including participants' own political orientations (1 = *strongly liberal*, 7 = *strongly conservative*). We will also ask for age, gender, education level, income, and race/ethnicity using the same measures as were used in the Ideology 2.0 study.

## II. Studies 1 and 2: The Ideology 2.0 dataset

**Open access to data and code.**   Across studies, we report how we determined sample size, how all participant exclusions were determined, and all manipulations and measures. All data used to derive this registered report and annotated R code (including proof-of-concept MSE estimates, which are described in the project OSF Wiki) are on the project's OSF page: https://osf.io/buwp7/?view_only=3905d5d1d54b4a499483c03a089b9f6e. Upon acceptance of the Stage 1 registered report, we will pre-register the studies and planned analyses and then we will be given the full data. The full data will be posted on the project's OSF page upon the final submission.

**Participants and procedure.**   We use data from the Ideology 2.0 study for both Study 1 and 2. Data from the Ideology 2.0 study were collected between December 2007 and June 2012. The data were collected from the Project Implicit website using a planned missingness design [25]. There were over 280,000 unique sessions, 40 reaction time measures (i.e., implicit measures), 30 self-report measures that matched the IAT targets, 25 individual difference questionnaires, and many individual self-report items. In each session, participants were randomly assigned 15 minutes' worth of items (they could complete multiple sessions if they chose to). Participants either completed one reaction time measure and nine explicit measures on the same topic or completed reaction time and explicit measures for two different topics. Topics included groups (our focus), but also specific concepts (e.g., fascism) and specific people (e.g., George W. Bush).

Planned missingness is a strategy used in data collection where participants are randomly assigned to respond to only certain items [28, 29]. The missing data points are missing completely at random (MCAR), so we can assume that the missing data will not systematically impact the results. That is, while each participant has missing data for certain variables, there is no consistent pattern across participants regarding which variables are missing. Therefore, the missingness can effectively be ignored in the analyses [30]. Other researchers who have collected large datasets (e.g., [31, 32]) have used MCAR data strategies to reduce the burden on participants and the cost of data collection.

The researchers from the Ideology 2.0 study made their exploratory data available to other researchers to use for registered report studies [25]. Currently, we have access to 22% of the data for exploratory purposes. In the initial stage, Schmidt and colleagues released

**Table 2. Demographics from the ideology 2.0 dataset.**

| Data Including IAT and Difference Score Explicit Measures | |
|---|---|
| N | 24296 |
| M age | 31.57 |
| SD age | 12.98 |
| % Men | 32.28 |
| % Women | 67.53 |
| % University degree | 46.22 |
| Data Including Only Single Target Explicit Ratings | |
| N | 6379 |
| M age | 31.65 |
| SD age | 12.98 |
| % Men | 33.27 |
| % Women | 66.53 |
| % University degree | 47.25 |

Table 2 contains anticipated demographics based on the Ideology 2.0 [25] masked data.

confirmatory *masked* data to researchers who requested it. From this confirmatory masked data, we have determined that there are 24,296 relevant sessions for this study (we filtered out sessions where participants did not respond to our measures of interest and where respondents were not from the United States). Table 2 contains anticipated demographics based on the masked data. Upon in-principle acceptance of our registered report, Schmidt and colleagues will give us access to their full data. The full description of the measures we use in this study is included below.

**Ideology 2.0 demographics measures used in Studies 1 and 2.** The Ideology 2.0 dataset included these demographic measures: political ideology, age, education, ethnicity, religion, gender, and income. Liberal/conservative ideology was measured such that 1 = *very liberal* and 7 = *very conservative*. Participants self-reported their age in years. Education was measured with five categories ranging from no high school diploma to graduate-level education. Ethnicity had nine categories: American Indian/Alaskan Native, East Asian, South Asian, Native Hawaiian or other Pacific Islander, White, Black or African American, and three categories describing some combination of these categories, or other. The religion variable initially contained 48 largely overlapping categories so we recoded it into Christian, Mormon, Jewish, Muslim, Hindu, Buddhist, and "other" categories. Gender had two categories, male and female. Income was included as a measure but there was substantial missing data on this measure, so we will omit it from our analyses.

## III. Study 1

**Absolute prejudice measures.** To measure absolute prejudice levels toward each target group, we will use three measures in the Ideology 2.0 dataset in Study 1. Participants responded to measures of the valence of their "gut" feelings about targets, their actual feelings toward targets, and how positively and negatively they felt about targets. The names of the items in the dataset, the number of complete cases available for each item, item wording, and scale endpoints are listed in Table 3. The groups for which this is available, and which will be included in analyses are shown in Table 4 with available cases for each.

**Analytic strategy—Predicting absolute prejudice using existing models.** With the measures of absolute prejudice as the outcome in the Ideology 2.0 data, we will use the same

**Table 3. Absolute measures of prejudice in the ideology 2.0 dataset.**

| | Items used to rate standalone targets (Sample 2) | | | |
|---|---|---|---|---|
| **Item** | **Wording** | **Scale endpoints** | **N** | **% Responded** |
| gut_x | What are your gut feelings toward x? | 7 = Strongly positive, 6 = Moderately positive, 5 = Slightly positive, 4 = Neither positive nor negative, 3 = Slightly negative, 2 = Moderately negative, 1 = Strongly negative | 3597 | 56.39 |
| gut_y | What are your gut feelings toward y? | 7 = Strongly positive, 6 = Moderately positive, 5 = Slightly positive, 4 = Neither positive nor negative, 3 = Slightly negative, 2 = Moderately negative, 1 = Strongly negative | 3597 | 56.39 |
| act_x | What are your actual feelings toward x? | 7 = Strongly positive, 6 = Moderately positive, 5 = Slightly positive, 4 = Neither positive nor negative, 3 = Slightly negative, 2 = Moderately negative, 1 = Strongly negative | 3596 | 56.37 |
| act_y | What are your actual feelings toward y? | 7 = Strongly positive, 6 = Moderately positive, 5 = Slightly positive, 4 = Neither positive nor negative, 3 = Slightly negative, 2 = Moderately negative, 1 = Strongly negative | 3596 | 56.37 |
| neg_x | Considering only the negative things about x and ignoring the positive things, how negative are those things? | 1 = Extremely negative, 2 = Very negative, 3 = Moderately negative, 4 = Slightly negative, 5 = Barely negative, 6 = not at all negative | 3627 | 56.86 |
| neg_y | Considering only the negative things about y and ignoring the positive things, how negative are those things? | 1 = Extremely negative, 2 = Very negative, 3 = Moderately negative, 4 = Slightly negative, 5 = Barely negative, 6 = not at all negative | 3629 | 56.89 |
| pos_x | Considering only the positive things about x and ignoring the negative things, how positive are those things? | 6 = Extremely positive, 5 = Very positive, 4 = Moderately positive, 3 = Slightly positive, 2 = Barely positive, 1 = not at all positive | 3628 | 56.87 |
| pos_y | Considering only the positive things about y and ignoring the negative things, how positive are those things? | 6 = Extremely positive, 5 = Very positive, 4 = Moderately positive, 3 = Slightly positive, 2 = Barely positive, 1 = not at all positive | 3633 | 56.95 |

Table 3 contains the names of the items in the dataset, the number of complete cases available for each item, item wording, and scale endpoints for the absolute measures of prejudice.

procedure for predicting ideology-prejudice associations that Brandt used [1]. We will use the R code from Brandt [1] as a guide to reproduce the models. His original code is available online: https://osf.io/g28yc/. First, we identify target groups in the Ideology 2.0 data (18 in this case, with nine from Brandt's [1] original study and nine new groups). These groups are described in Table 4 above.

**Power analyses for the models looking at the absolute measures.** We used the InteractionPowerR Shiny App for analytic power [33] to examine our ability to detect the relationship between perceived target characteristics and the ideology-prejudice relationship for the absolute measures. The app calculates power based on the anticipated correlations between predictors, predictors and the DV, and the strength of the interaction to be tested (x1*x2 and y). For the absolute measures with our final anticipated sample size of 6379, the analyses suggested that we have an 90% chance of detecting an effect with correlations between our predictors as well as each predictor and the outcome slope as .10, and an interaction term of .05. This suggests we are well-powered to detect small effects.

**Calculating predicted and observed prejudice values.** To examine whether Brandt's [1] models are robust, we will input the prejudice values from the Ideology 2.0 dataset [25] as well as the values for perceptions of group ideology, status, and choice from our newly collected data into his equations (see Table 1), with a separate equation for each group. All the equations for the predicted values are in the code on the project's OSF page, and the OSF Wiki provides a guide as to where each component resides in the file. Then, we will solve the predictive equations to generate estimates of the ideology-prejudice association for each target group in the

**Table 4. Number of absolute measure responses per distinct group in the data.**

| Groups from current study | In Brandt (2017) Study? | Number of Participants Available |
|---|---|---|
| *Absolute Measures* | | |
| Gay | Yes | 731 |
| Straight | **No** | 731 |
| Democrats | Yes | 757 |
| Republicans | Yes | 757 |
| Liberals | Yes | 738 |
| Conservatives | Yes | 738 |
| Non-Profits | **No** | 690 |
| Corporations | **No** | 690 |
| Labor | **No** | 644 |
| Management | **No** | 644 |
| Foreign | **No** | 692 |
| Local | **No** | 692 |
| Black | Yes | 731 |
| White | Yes | 731 |
| Religious | Yes | 768 |
| Atheist | Yes | 768 |
| Mother | **No** | 738 |
| Father | **No** | 738 |

Table 4 contains the number of participants per group for the absolute measures.

context of each of the four models. We call these estimates the predicted ideology-prejudice association. To do this, we will regress each measure of prejudice for each target group on ideology and demographic control variables [34]. In these analyses we will use American Indian/Alaskan Native as the reference category for ethnicity, no high-school diploma for the education reference category, Christian as the religion reference category, and male as the gender reference category, and mean centered age. The measures of prejudice and ideology will be rescaled to range from 0 to 1. The estimate for ideology from this model is our observed ideology-prejudice estimate for each target group and each measure of prejudice.

For example, referring to Table 1, to calculate the ideology-prejudice association for Black people using the ideology-only model (and using Brandt's [1] original equation and the group ratings from Brandt [1]), we would use this equation, $\hat{y} = 0.022 - 1.420(\text{ideology})$, and substitute "(ideology)" with 31.7: $\hat{y} = 0.022 - 1.420(31.7)$, to arrive at the estimate, -44.99.

Once we have these estimates, we will estimate the relationship between ideology and absolute prejudice against each group. We call these estimates the observed ideology-prejudice association. We will examine how well the predicted association from each of our four models maps onto the observed association for each target group in the Ideology 2.0 data. We will estimate the mean squared error (MSE) of the observed ideology-prejudice association compared to the predicted association. MSEs serve as a means of evaluating the accuracy of a predictive model by comparing predicted and actual values. To calculate MSEs, we will take the actual ratings of prejudice across targets, subtract them from the estimates from the equations, square these values, and then take the average. Then, we will run a mixed ANOVA and post-hoc tests using the jmv package in R [35] to determine whether the differences between model MSEs are statistically significant (using an alpha level of.05). We will opt to use the original *p* values for the post-hoc tests as opposed to a procedure like Bonferroni or Tukey because we do not wish to omit potentially meaningful significant results based on stringent adjustments [36].

The model that has the lowest MSE (and the difference between that model and the next-lowest model is statistically significant) can be considered the most accurate predictive model of prejudice in this context. The calculations for the MSEs as well as the mixed ANOVA with post-hoc tests are included in the code on the project's OSF page. The OSF Wiki contains details related to which lines of the code correspond to each calculation and test.

**Comparing measure performance.** As part of our proof-of-concept analyses, we looked at how well the predictive estimates for each model mapped onto the analyses for each absolute measure of prejudice. We also compared how well each model performed compared to the other models. This is an incomplete analysis because we do not have perceived group characteristics for all the target groups we plan to study. For our analyses, we first calculated the MSEs for all absolute prejudice measures across all available groups for each model. Then, we calculated MSEs for each absolute measure separately for each model. We estimated a mixed ANOVA with each of the five types of prejudice measures (all measures in combination, and each of the four individual prejudice measures) as a within-subjects factor and model type as a between-subjects factor with five levels to test if model accuracy depends on the models and type of measures.

For these proof-of-concept analyses, we focus on the effects of model and the effect of type of prejudice measure. We ran these models using the nine target groups that we currently have ratings for (Black people, White people, gay people, liberals, conservatives, Democrats, Republicans, religious people, atheists). We cannot run the models using all 18 target groups here because characteristic ratings may have changed since previous data were collected and because some groups in the Ideology 2.0 dataset are not included in Brandt's [1] group rating data.

Importantly, however, these proof-of-concept analyses were run using the same code that we will use for the final analyses. In the final analyses, we will have a more complete idea of the possible interactive effects between model type and type of prejudice measure because we will be able to run analyses using the full data (as opposed to the subset we use here). Using an alpha level of .05, the effect of type of prejudice measure was significant, $F(4,160) = 14.98$, $p < .001$. This indicates that the predictive equations are more accurate at predicting some measures of prejudice than others. There was a significant effect of model type, $F(4,40) = 2.81$, $p = .038$. This indicates that some models had greater predictive accuracy than others. Additionally, there was a significant interaction between outcome measure and model type, $F(16,160) = 3.62$, $p < .001$. This indicates that whether a model was accurate or not depended on the outcome measure.

The mean differences for the main effect of the type of prejudice measure are reported in Table 5. The average of all measures was better at predicting the ideology-prejudice association than the single measures for actual and gut feelings. The positive prejudice measure performed better than the average of all measures and better than the single actual and gut feeling measures. The negative measure of prejudice performed better than the average of all measures as well as the single actual and gut feeling measures.

The mean differences for the main effect of model type are reported in Table 6. Overall, the ideology-only model was the best performer, as it had a significantly lower MSE than all other models except for the combined ideology, status, and choice model. The ideology, status, and choice model performed better than the choice-only and null models.

There was a significant interaction between model and measure type. In the final version of the manuscript, we will calculate and interpret these differences. However, for the purposes of this proof-of-concept analysis, we are not convinced that these are good tests given that we have fewer-than-planned target groups and smaller-than-planned sample sizes. For the curious

**Table 5.  Post hoc tests for absolute prejudice measures.**

| Comparison | | | | | |
|---|---|---|---|---|---|
| Prejudice Measures | Mean Difference | SE | df | t | p |
| All Measures—Actual | -0.04 | 0.01 | 40 | -4.04 | < .001 |
| All Measures—Gut | -0.05 | 0.01 | 40 | -5.05 | < .001 |
| All Measures—Positive | 0.03 | 0.01 | 40 | 2.66 | .011 |
| All Measures—Negative | 0.03 | 0.01 | 40 | 3.50 | .001 |
| Actual—Gut | -0.01 | 0.01 | 40 | -1.58 | .122 |
| Actual—Positive | 0.07 | 0.02 | 40 | 3.66 | < .001 |
| Actual—Negative | 0.07 | 0.02 | 40 | 4.15 | < .001 |
| Gut—Positive | 0.08 | 0.02 | 40 | 3.92 | < .001 |
| Gut—Negative | 0.08 | 0.02 | 40 | 4.81 | < .001 |
| Positive—Negative | 0.001 | 0.01 | 40 | 0.13 | .896 |

Table 5 contains the mean differences for the main effect of the type of absolute prejudice measure.

reader, Table 1 includes the MSEs for each model and for each type of measure from Brandt's [1] original study.

It is important to note that we only have access to a smaller subset of the exploratory data, and more nuances may emerge in the full data. Therefore, these results represent a proof-of-concept and demonstrate that we can run the relevant models on the full data. We have included Fig 1 as an example to show how we will plot the results of each model for each group in the Stage 2 report. At that stage, we will plot the results for each model for each group, per prejudice outcome measure.

## IV. Study 2

**Explicit relative prejudice measures.**   Explicit relative prejudice in the Ideology 2.0 dataset was measured in five ways. First, participants were asked to indicate which group they preferred when given a choice between two groups (e.g., Black people and White people). Using the "gut," "actual," "positive," and "negative" feelings measures described above, difference scores were created for each respective measure to determine preferences for some target

**Table 6.  Post hoc tests for model comparisons.**

| Comparison | | | | | |
|---|---|---|---|---|---|
| Model | Mean Difference | SE | df | t | p |
| Ideology—Status | -0.13 | 0.06 | 40 | -2.09 | .043 |
| Ideology—Choice | -0.14 | 0.06 | 40 | -2.34 | .024 |
| Ideology—Ideology + Status + Choice | -0.01 | 0.06 | 40 | -0.21 | .837 |
| Ideology—Null | -0.14 | 0.06 | 40 | -2.34 | .024 |
| Status—Choice | -0.01 | 0.06 | 40 | -0.25 | .805 |
| Status—Ideology + Status + Choice | 0.11 | 0.06 | 40 | 1.89 | .067 |
| Status—Null | -0.01 | 0.06 | 40 | -0.25 | .805 |
| Choice—Ideology + Status + Choice | 0.13 | 0.06 | 40 | 2.13 | .039 |
| Choice—Null | < .001 | 0.06 | 40 | < .001 | .999 |
| Ideology + Status + Choice—Null | -0.13 | 0.06 | 40 | -2.13 | .039 |

Table 6 contains the mean differences for the main effect of model type.

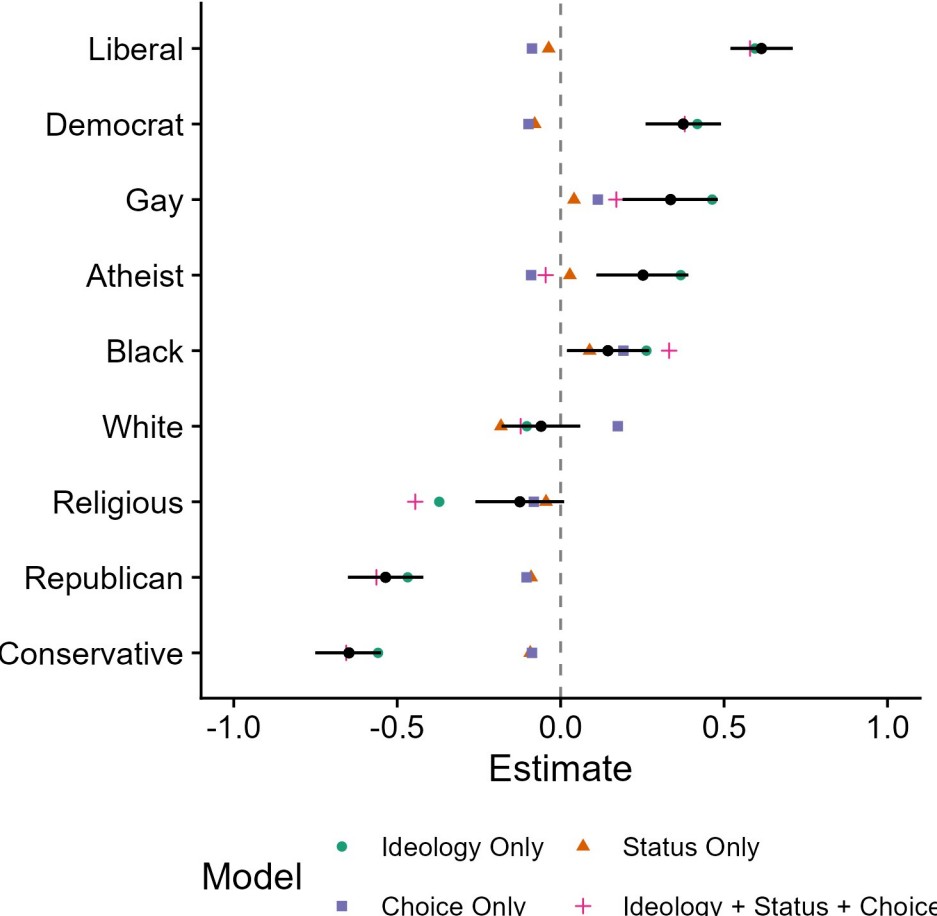

**Fig 1. Depiction of actual and model estimated slopes.** Error bars are 95% confidence interval.

groups over others. The "gut difference" measure was calculated by subtracting the gut feeling score for group Y minus the score for group X (e.g., where White people are group Y and Black people are group X). The "actual difference" measure was calculated by subtracting the actual feeling score for group Y minus the score for group X. The "positive difference" measure was calculated by subtracting the positive feeling score for group Y minus the score for group X. The "negative difference" measure was calculated by subtracting the negative feeling score for group Y minus the score for group X. In all cases except one, the more traditionally liberal group is coded as group X (e.g., gay people). The one case where the more traditionally conservative group is coded as group X is the religious people vs. atheists pairing. In this case, we reversed the score for this group comparison. The number of participants who responded to each measure is included in Table 7. The groups for which this is available, and which will be included in analyses are shown in Table 8 with available cases for each.

**Reaction time outcomes.** When completing the IAT [18], participants pair groups (i.e., Black and White) with positive and negative stimuli (i.e., "good" and "bad" words). The outcome measure of the IAT is a reaction time measure, which is indicated by a D score [18]. This score indicates a preference for one group over another such that a faster reaction time when pairing one group with positive stimuli (as opposed to negative stimuli) indicates a preference for that group. In this dataset, D scores are coded in the same way as the differences between

**Table 7. Measures of explicit relative prejudice plus IAT D score in the ideology 2.0 dataset.**

| Items used in the data including IAT responses (Sample 1) | | | |
|---|---|---|---|
| **Item** | **Wording** | **N** | **% Responded** |
| pref_xy | Which group do you prefer, x or y | 24249 | 99.81 |
| gut_diff | Gut feelings toward x—gut Feelings toward y | 3594 | 14.79 |
| act_diff | Actual feelings toward x—actual feelings Toward y | 3594 | 14.79 |
| neg_diff | Negative feelings toward x—negative feelings toward y | 3620 | 14.90 |
| pos_diff | Positive feelings toward x—positive feelings toward y. | 3622 | 14.90 |
| D_score_xy | IAT D-scores in same direction as explicit measure scoring | 9643 | 39.69 |

Table 7 contains the number of participants who responded to each type of explicit relative prejudice measure. The last row represents the IAT D score.

explicit measures (see the Target X and Target Y columns in Table 9). We use the IAT as a measure of comparative prejudice. The number of participants who completed IAT tasks are listed in Table 7.

**Power analyses for the comparative measures.** We used the InteractionPowerR Shiny App for analytic power [33] to examine our ability to detect the relationship between perceived target characteristics and the ideology-prejudice relationship. For the comparative measures using our final anticipated sample size of 24296, the analyses suggested that we have an 82% chance of detecting an effect with correlations between our predictors as well as each predictor and the outcome slope as .06, and an interaction term of .02. This suggests we are well-powered to detect small effects.

**Analytic strategy—Creating new models to predict comparative prejudice.** In Study 2, we will create new predictive multilevel models using the lme4 package in R [34], using the Ideology 2.0 data with the goal of examining which perceived aspects of target groups are most predictive of the comparative ideology-prejudice relationship. To do this, we will split our data into two parts (80/20 split) at random for each target group pair [26] and run four conceptually identical multilevel models to those used for the ideology-absolute prejudice models. Typically, in the train-test technique, the training set includes 60–80% of the available data and the test set includes 20–40% (e.g., [11, 37]). The train-test method allows us to do three things: assess overall ideological-prejudice association prediction accuracy, measure prediction accuracy using a new sample, and incorporate a prediction model that is designed specifically for our

**Table 8. Number of relative measure responses per distinct group in the data.**

| Groups from current study | In Brandt (2017) Study? | Number of Participants Available |
|---|---|---|
| | **Relative Measures** | |
| Black vs. White | Yes | 2748 |
| Gay vs. Straight | **Straight only** | 2869 |
| Mother vs. Father | **No** | 2777 |
| Foreign vs. Local | **No** | 2659 |
| Labor vs. Management | **No** | 2543 |
| Non-Profits vs. Corporations | Yes | 2608 |
| Democrat vs. Republican | Yes | 2631 |
| Liberal vs. Conservative | Yes | 2634 |
| Religious vs. Atheist | Yes | 2827 |

Table 8 contains the number of participants per group for the relative measures.

**Table 9. IAT tasks.**

| Target X | Target Y | Preference IAT Available |
|---|---|---|
| Gay | Straight | Evaluation (Good/ Bad) |
| Non-profits | Corporations | Evaluation (Good/ Bad) |
| Labor | Management | Evaluation (Good/ Bad) |
| Foreign | Local | Evaluation (Good/ Bad) |
| Black | White | Evaluation (Good/ Bad) |
| Mother | Father | Evaluation (Good/ Bad) |

We will not use reaction time measures from IATs comparing a group to the self (i.e., Democrat/Republican, Liberal/ Conservative, and Religious/Atheist). We will use the evaluative IATs comparing groups in terms of positive and negative valence (i.e., good/bad).

research questions [26]. We will use 80% of the data for training the models, and 20% for testing the models. The data will be split for each unique target group pair. The code for the multilevel models is included in the code on the project's OSF page. The OSF Wiki contains details related to which lines of the code correspond to each calculation and test.

We will run models that parallel the models in Study 1. In the ideology-only model, we will include our measure of the difference in perceived ideology for the groups in the pairs predicting how well participant ideology explains the difference in prejudice between each target in a pair (e.g., Black vs. White people). Additional models will use perceived differences in status and choice respectively to predict the relationship between ideology and comparative prejudice for each group, and a fourth model will use perceived differences in ideology, status, and choice in combination to predict the relationship between participant ideology and comparative prejudice.

**Training the data.** We will use the training set data to create our predictive models of comparative prejudice. To examine measures of comparative prejudice against target groups we will estimate multilevel models using the lme4 package in R [34] on the first random 80% of the Ideology 2.0 data. In the models, we will nest the measures of comparative prejudice against different target group pairs within participants and include random intercepts to account for overall differences between participants and target pairs in terms of overall levels of comparative prejudice. We will control for gender with men as the reference group, education with less than a high school education as the reference group, religion with Christians as the reference group, ethnicity with American Indian/Alaskan Native as the reference group and mean-centered age. All variables will be recoded to range from 0–1 so that the coefficients can be interpreted as the percent of change in the outcome as one goes from the lowest to the highest value in the measure.

In the models, we will regress comparative prejudice on participant ideology, subsets of the group characteristics, and their interaction. To create our five predictive models, we will include perceived differences in ideology, status, choice, and all three measures simultaneously as predictors of the ideology-comparative prejudice relationship. These will be treated as random slopes [38]. These slopes represent the interaction terms between the relationship between participant ideology and comparative prejudice and each perceived target group characteristic. This will allow us to examine the amount that each of these perceived traits impacts the relationship between participant ideology and comparative prejudice.

Once we have built our predictive models, we will run multilevel models using the lme4 package in R [34] using the training data set. We will enter the relevant values for ideology, status, and choice into the equations to generate an estimated level of comparative prejudice for

each target pair using each of our four models. This will give us an estimated level of comparative prejudice in each instance that we can compare to the observed values in the data. For the sake of demonstration, here is Brandt's [1] status-only model: $\hat{y} = 0.001 - 0.846(status)$. If we wanted to estimate the ideology-prejudice association for atheists, we would input the status rating for atheists. We will generate our own value for this once we collect our pilot data, but for this example, we will use the rating from Brandt [1], 48.3. We will insert the status value into the equation, $\hat{y} = 0.001 - 0.846(48.3)$ to arrive at the predicted value, -40.86.

**Testing model fit.** After we have trained our models using the training data, we will use the test data (the remaining 20% of the Ideology 2.0 dataset, for each target group) to estimate the observed relationship between ideology and comparative prejudice for each target group pair and see which predictive model captures observed comparative prejudice most closely. We will do this by estimating the MSE of the observed ideology-comparative prejudice association compared to the predicted association. By comparing the MSEs of each of the five models predicting the participant ideology-prejudice relationship across targets and the observed values of prejudice against each target in the test set, we can test which model(s) are most effective in predicting prejudice across targets. Lower MSEs when comparing the slopes in the estimates to the observed levels of participant ideology-comparative prejudice indicate better fit in this context.

It is important to note that we are unable to test our models for the relative measures without the pilot group perceptions data (because nine of the groups were not included in Brandt's original study).

**Projected results.** We anticipate that the models will perform similarly here as they performed in Brandt's [1] original study, such that there will be at least one model that is more predictive of the ideology-prejudice association than the others. We expect that Model 1 (ideology-only) will be the best-performing model in terms of accuracy and parsimony.

## Implications

This work represents a stress-test of the models proposed by Brandt [1] using novel target groups, reaction time measures, and alternative explicit measures. In Study 1, we will directly replicate Brandt [1] under a different context. This will provide insight into the predictive power of these models. If the original findings are replicated, this will indicate that these models validly predict the ideology-prejudice association across different contexts. The predictive power of these models could be used to inform future hypotheses regarding the direction and magnitude of the ideology-prejudice association for specific target groups and allow researchers to make specific predictions of effect size, which would allow for a stringent test of their hypotheses.

In Study 2, we extend Brandt's [1] original work by developing new models using a different dataset from the one he used. If our results are similar, this will indicate that this strategy of model-building is effective across contexts. This strategy can be used in future research when attempting to predict other relationships outside the prejudice literature. The train/test procedure is becoming more common in psychology, and if successful, this work will contribute to the growing body of model prediction literature.

Additionally, this study could provide insight into the predictive power of not just absolute, but relative measures of prejudice, providing a useful advancement when comparing feelings toward some groups compared to others. These new relative predictive models may also be useful to researchers who perform reaction time work, as no current predictive models to our knowledge have been built to accommodate comparative study designs. In addition, the current work will also allow us to examine whether or not the predictive power of both absolute

and relative measures of prejudice varies between models, groups, and outcome measures, providing a more nuanced context in which to predict ideological prejudice.

## Supporting information

**S1 Appendix. New data collection survey questions.** This supplemental file includes feeling thermometers, status rating, ideology rating, and choice rating items for the new data collection described in the manuscript.
(DOCX)

## Acknowledgments

We would like to thank Kathleen Schmidt and the Ideology 2.0 team for putting out the call for registered reports and for allowing us to use their data.

## Author Contributions

**Conceptualization:** Jordan L. Thompson, Mark J. Brandt, Geoffrey A. Wetherell.

**Formal analysis:** Jordan L. Thompson, Mark J. Brandt, Geoffrey A. Wetherell.

**Funding acquisition:** Mark J. Brandt, Geoffrey A. Wetherell.

**Investigation:** Jordan L. Thompson, Mark J. Brandt, Geoffrey A. Wetherell.

**Methodology:** Jordan L. Thompson, Mark J. Brandt, Geoffrey A. Wetherell.

**Project administration:** Jordan L. Thompson.

**Resources:** Jordan L. Thompson, Abigail L. Cassario, Mark J. Brandt, Geoffrey A. Wetherell.

**Software:** Jordan L. Thompson, Sada Rice, Mark J. Brandt, Geoffrey A. Wetherell.

**Supervision:** Mark J. Brandt, Geoffrey A. Wetherell.

**Validation:** Jordan L. Thompson, Mark J. Brandt, Geoffrey A. Wetherell.

**Visualization:** Jordan L. Thompson, Mark J. Brandt, Geoffrey A. Wetherell.

**Writing – original draft:** Jordan L. Thompson.

**Writing – review & editing:** Jordan L. Thompson, Abigail L. Cassario, Shree Vallabha, Samantha A. Gnall, Sada Rice, Prachi Solanki, Alejandro Carrillo, Mark J. Brandt, Geoffrey A. Wetherell.

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
