## [Decision Letter · Decision Letter 0]

29 Sep 2023

PONE-D-23-11879Registered report protocol: Stress testing predictive models of ideological prejudicePLOS ONE

Dear Dr. Thompson,

Thank you for submitting your manuscript to PLOS ONE. After careful consideration, we feel that it has merit but does not fully meet PLOS ONE’s publication criteria as it currently stands. Therefore, we invite you to submit a revised version of the manuscript that addresses the points raised during the review process.

ACADEMIC EDITOR: Please see reviewers' comments below.

We look forward to receiving your revised manuscript.

Kind regards,

Anastassia Zabrodskaja, Ph.D.

Academic Editor

PLOS ONE

Journal Requirements:

3. We note that the original protocol that you have uploaded as a Supporting Information file contains an institutional logo. As this logo is likely copyrighted, we ask that you please remove it from this file and upload an updated version upon resubmission.

Reviewers' comments:

Reviewer's Responses to Questions

**Comments to the Author**

1. Does the manuscript provide a valid rationale for the proposed study, with clearly identified and justified research questions?

Reviewer #1: Partly

2. Is the protocol technically sound and planned in a manner that will lead to a meaningful outcome and allow testing the stated hypotheses?

Reviewer #1: Yes

3. Is the methodology feasible and described in sufficient detail to allow the work to be replicable?

Reviewer #1: No

4. Have the authors described where all data underlying the findings will be made available when the study is complete?

Reviewer #1: Yes

5. Is the manuscript presented in an intelligible fashion and written in standard English?

Reviewer #1: Yes

6. Review Comments to the Author

You may also provide optional suggestions and comments to authors that they might find helpful in planning their study.

Reviewer #1: 1.Original topic theme of already established work of Brandt (2017)

2. Need to explain more how the planned missing design works for exploratory data. Which softwares are used for post hoc? SPSS?

3. Although methodology is of high technical standard but it is underused in paper publications, need more latest citations.

4. Ethical standards are met. Sample is gathered online hence some evidence is good to share of how it has been collated. Also MSE calculated, method needs to be discussed and if calculations are present in repository.

7. PLOS authors have the option to publish the peer review history of their article (what does this mean?). If published, this will include your full peer review and any attached files.

Reviewer #1: **Yes: **yumna ali

---

## [Author Response · Author response to Decision Letter 0]

8 Jan 2024

Below, we have copied and pasted the editor's and reviewer's comments. Our responses are below each comment. We have also included our responses in our cover letter document.

Journal Requirements: 

Thank you for this helpful reminder. We have gone through PLOS ONE’s style requirements and updated our manuscript accordingly. 

Thank you for the clarification, we fully intend to provide repository information for our data at acceptance.

3. We note that the original protocol that you have uploaded as a Supporting Information file contains an institutional logo. As this logo is likely copyrighted, we ask that you please remove it from this file and upload an updated version upon resubmission.

Thank you for bringing this to our attention. We have updated the file so that the logo is removed, and the document is blinded for review. 

Reviewer 1

1. Does the manuscript provide a valid rationale for the proposed study, with clearly identified and justified research questions? The research question outlined is expected to address a valid academic problem or topic and contribute to the base of knowledge in the field. 

Reviewer #1: Partly

We thank the reviewer for this comment. As we edited the paper, we have made efforts to clarify our rationale for the proposed study. See pages 4 and pages 7-8 for two key examples of these changes:

(from page 4) “The prejudice and ideology literatures do not offer many concrete methodological suggestions for predicting the magnitude and direction of the effect of ideology on prejudice against specific groups. As an example, widely cited dual process models of prejudice suggest that a desire for social conformity and belief in a dangerous world predict right-wing authoritarian (RWA) attitudes, whereas the belief the world is a competitive jungle predicts social dominance, both of which predict prejudice [14]. Despite providing a theoretical rationale for the relationship between manifestations of ideology (i.e., RWA) and prejudice towards certain groups, this perspective does not provide scaffolding to predict the specific level of prejudice a person will exhibit towards a variety of target groups, in addition to examining a somewhat limited set of targets. This is a gap that should be filled because there is both practical and theoretical value in models that can predict the ideology-prejudice association across targets [4].”

(from pages 7-8) “The purpose of this project is to stress-test Brandt’s [4] original models using alternative measures of prejudice, including alternative explicit measures (i.e., relative measures of explicit prejudice), and reaction time measures (i.e., Implicit Association Test scores [20]). This addresses a key shortcoming of the original study. In particular, the original study used only feeling thermometers as an explicit measure of prejudice. Although feeling thermometers are common measures of prejudice (e.g., [10], [36], [18]), they are just one possible measure of group-based attitudes. The predictive accuracy of the models might be limited to feeling thermometers. If so, this would limit the utility of the predictive models to only studies that use feeling thermometers. This is a potentially substantial limitation given the diversity of prejudice measures that exist. However, if the predictive models work well with alternative measures of prejudice, i.e., the explicit and reaction time measures, it suggests that the model is much more generalizable. This would mean that this model could be used for different types of prejudice measures.”

2. Is the protocol technically sound and planned in a manner that will lead to a meaningful outcome and allow testing the stated hypotheses? The manuscript should describe the methods in sufficient detail to prevent undisclosed flexibility in the experimental procedure or analysis pipeline, including sufficient outcome-neutral conditions (e.g. necessary controls, absence of floor or ceiling effects) to test the proposed hypotheses and a statistical power analysis where applicable. As there may be aspects of the methodology and analysis which can only be refined once the work is undertaken, authors should outline potential assumptions and explicitly describe what aspects of the proposed analyses, if any, are exploratory. 

Reviewer #1: Yes

We appreciate the reviewer’s positive feedback regarding our work.

3. Is the methodology feasible and described in sufficient detail to allow the work to be replicable? 

Reviewer #1: No

We thank the reviewer for bringing this to our attention. Below where we respond to the reviewer’s verbal comments, we address this point in detail.

4. Have the authors described where all data underlying the findings will be made available when the study is complete? The PLOS Data policy requires authors to make all data underlying the findings described in their manuscript fully available without restriction, with rare exception, at the time of publication. The data should be provided as part of the manuscript or its supporting information, or deposited to a public repository. For example, in addition to summary statistics, the data points behind means, medians and variance measures should be available. If there are restrictions on publicly sharing data—e.g. participant privacy or use of data from a third party—those must be specified. 

Reviewer #1: Yes

We appreciate the reviewer’s positive feedback regarding our work.

5. Is the manuscript presented in an intelligible fashion and written in standard English? PLOS ONE does not copyedit accepted manuscripts, so the language in submitted articles must be clear, correct, and unambiguous. Any typographical or grammatical errors should be corrected at revision, so please note any specific errors here. 

Reviewer #1: Yes

We appreciate the reviewer’s positive feedback regarding our work.

1.Original topic theme of already established work of Brandt (2017)

We thank the reviewer for their reminder to clarify the original purpose of Brandt (2017). We have done this on page 3 and 9-10 of the manuscript. For example, the quotes below illustrate our intentions to replicate and extend his work.

We reiterated the original purpose of the study on page 3: 

“Brandt sought to predict the ideology-prejudice association for multiple target groups with models using ideology, perceived group status, and perceived choice of group membership as predictors [4]. The goal was to create models that make accurate predictions of the ideology-prejudice association toward a variety of target groups and to help us understand which theoretical perspectives are most useful when predicting ideological prejudice. We replicate and extend this work here.”

In addition, to provide additional context for our studies, we have added language about how our work will replicate and extend the original work on pages 9-10:

“We will replicate (Study 1) and extend (Study 2) Brandt’s [4] work in two ways. First, in Study 1, we will use models derived from the original work to estimate the extent to which the ideology-absolute-prejudice relationships for each group in the Ideology 2.0 data are explained by the models. If the original models are robust, they should predict the ideology-absolute-prejudice relationship in the Ideology 2.0 dataset with different measures with relative accuracy. If they are not predictive of the ideology-prejudice association measured using alternative explicit measures of absolute prejudice, it will provide valuable insight into how ideology may relate to different types of prejudice and the boundary conditions of the models. 

Study 2 goes beyond this. In this study, we will build new predictive models of relative prejudice. This contributes to the literature by adding conceptual and computational depth to existing work. We examine the predictive abilities of our models by leveraging the size of the Ideology 2.0 dataset to perform a train/test split [29].”

2. Need to explain more how the planned missing design works for exploratory data. Which softwares are used for post hoc? SPSS?

We thank the reviewer for this request for additional clarity regarding planned missingness designs. We have added clarifying language regarding the strategy behind using missing completely at random (MCAR) designs on pages 15-16 (see quote below). 

“Planned missingness is a strategy used in data collection where participants are randomly assigned to respond to only certain items ([17], [27]). The missing data points are missing completely at random (MCAR), so we can assume that the missing data will not systematically impact the results. That is, while each participant has missing data for certain variables, there is no consistent pattern across participants regarding which variables are missing. Therefore, the missingness can effectively be ignored in the analyses [26]. Other researchers who have collected large datasets (e.g., [9], [28]) have used MCAR data strategies to reduce the burden on participants and the cost of data collection.”

Additionally, we thank the reviewer for the request for clarification regarding post hoc tests and statistical software. We have added a clearer description of our strategy for our post hoc tests on page 21 (see quote below). For clarity, all analyses will be run using R Studio.

“Then, we will run a mixed ANOVA and post-hoc tests using the jmv package in R [30] to determine whether the differences between model MSEs are statistically significant (using an alpha level of .05). The model that has the lowest MSE (and the difference between that model and the next-lowest model is statistically significant) can be considered the most accurate predictive model of prejudice in this context. The calculations for the MSEs as well as the mixed ANOVA with post-hoc tests are included in the code on the project’s OSF page. The OSF Wiki contains details related to which lines of the code correspond to each calculation and test.”

Furthermore, we have added specific details about how MSEs are calculated on page 21 for the absolute measures (see quote below): 

“We will estimate the mean squared error (MSE) of the observed ideology-prejudice association compared to the predicted association. MSEs serve as a means of evaluating the accuracy of a predictive model by comparing predicted and actual values. To calculate the MSEs, we will take the actual ratings of prejudice across targets, subtract them from the estimates from the equations, square these values, and then take the average.”

The MSE calculations for the relative measures are described on page 31 (see quote below):

“After we have trained our models using the training data, we will use the test data (the remaining 20% of the Ideology 2.0 dataset, for each target group) to estimate the observed relationship between ideology and comparative prejudice for each target group pair and see which predictive model captures observed comparative prejudice most closely. We will do this by estimating the MSE of the observed ideology-comparative prejudice association compared to the predicted association. By comparing the MSEs of each of the five models predicting the participant ideology-prejudice relationship across targets and the observed values of prejudice against each target in the test set, we can test which model(s) are most effective in predicting prejudice across targets. Lower MSEs when comparing the slopes in the estimates to the observed levels of participant ideology-comparative prejudice indicate better fit in this context.”

3. Although methodology is of high technical standard but it is underused in paper publications, need more latest citations.

We thank the reviewer for bringing up this point. We have now highlighted the advantages of our methodology on page 5, and we have cited more recent publications that use similar techniques (see quote below).

“Likewise, predictive models have theoretical implications because predictions of effect size can be useful in falsifying hypotheses and comparing rival models. Indeed, recent research in psychology has used predictive models to predict diverse outcomes, such as changes in trust [33], romantic interest [15], and success in psychotherapy [35]. Here we further develop predictive models of the ideology-prejudice association.”

1. Rosenbusch H, Soldner F, Evans AM, Zeelenberg M. Supervised machine learning methods in psychology: A practical introduction with annotated R code. Soc Pers Psychol Compass. 2021;15(2):e12579. doi: 10.1111/spc3.12579

2. Lanning K, Wetherell G, Warfel EA, Boyd RL. Changing channels? A comparison of Fox and MSNBC in 2012, 2016, and 2020. Anal Soc Issues Public Policy. 2021;21(1):149-174. doi: 10.1111/asap.12265

3. Stavrova O, Evans AM, Brandt MJ. Ecological dimensions explain the past but do not predict future changes in trust. Am Psychol. 2021;76(6):983. doi: 10.1037/amp0000815

4. Eastwick PW, Joel S, Carswell KL, Molden DC, Finkel EJ, Blozis SA. Predicting romantic interest during early relationship development: A preregistered investigation using machine learning. Eur J Pers. 2023;37(3):276-312. doi: 10.1177/08902070221085877

4. Ethical standards are met. Sample is gathered online hence some evidence is good to share of how it has been collated. Also MSE calculated, method needs to be discussed and if calculations are present in repository.

We thank the reviewer for pointing out the lack of clarity regarding data collection. To clarify, we have moved language to the section on IRB approval of the original study to pages 11-12 of the manuscript. 

“Participants in the Ideology 2.0 study participated in an online study. They read a consent document and advanced to the screen with the study instructions only if they agreed to participate. Because of the nature of online studies, it was not possible to obtain written consent as any signatures obtained would inevitably be linked to specific participants. The researchers who own the dataset obtained approval from the University of Virginia’s Institutional Review Board for this procedure.

In the new data collection, the consent document will precede the survey. Participants can read about the study and decide whether they would like to participate. If participants choose to participate, they can click an arrow button to advance to the survey (which will indicate informed consent). If someone chooses not to participate, they can close out of the survey with no penalty. The current protocol involves online data collection and does not contain any identifying or sensitive information. It is not possible for participants to provide a signature in this context and doing so would make participant data identifiable given that the signature would be linked to their survey responses. The Institutional Review Board at BLINDED FOR REVIEW University approved this procedure.”

We hope that this new placement will augment the language on page 15 describing the Project Implicit study design: 

“We use data from the Ideology 2.0 study for both Study 1 and 2. Data from the Ideology 2.0 study were collected between December 2007 and June 2012. The data were collected from the Project Implicit website using a planned missingness design [31]. There were over 280,000 unique sessions, 40 reaction time measures (i.e., implicit measures), 30

---

## [Decision Letter · Decision Letter 1]

13 Feb 2024

PONE-D-23-11879R1Registered report: Stress testing predictive models of ideological prejudicePLOS ONE

Dear Dr. Thompson,

Thank you for submitting your manuscript to PLOS ONE. After careful consideration, we feel that it has merit but does not fully meet PLOS ONE’s publication criteria as it currently stands. Therefore, we invite you to submit a revised version of the manuscript that addresses the points raised during the review process.

We look forward to receiving your revised manuscript.

Kind regards,

Anastassia Zabrodskaja, Ph.D.

Academic Editor

PLOS ONE

Reviewers' comments:

Reviewer's Responses to Questions

**Comments to the Author**

1. Does the manuscript provide a valid rationale for the proposed study, with clearly identified and justified research questions?

Reviewer #1: Partly

2. Is the protocol technically sound and planned in a manner that will lead to a meaningful outcome and allow testing the stated hypotheses?

Reviewer #1: Partly

3. Is the methodology feasible and described in sufficient detail to allow the work to be replicable?

Reviewer #1: No

4. Have the authors described where all data underlying the findings will be made available when the study is complete?

Reviewer #1: No

5. Is the manuscript presented in an intelligible fashion and written in standard English?

Reviewer #1: Yes

6. Review Comments to the Author

You may also provide optional suggestions and comments to authors that they might find helpful in planning their study.

Reviewer #1: Dear authors,

The study is unque and prejudice is integral to practically study, However

-Methodolgy is tough to interpret. How can data be accessed as it is gathered in stages.

-Each data analytic step has not been meaningul explained e.g post-hoc tests were used by Turkey or Bonferroni?

-If data is not yet full or reached limit then must be waited for the complete findings.

7. PLOS authors have the option to publish the peer review history of their article (what does this mean?). If published, this will include your full peer review and any attached files.

Reviewer #1: **Yes: **yumna ali

---

## [Author Response · Author response to Decision Letter 1]

7 Apr 2024

We thank the editor and reviewer for their time, consideration, and valuable comments when reviewing our work. Below are our comments regarding the current incarnation of the paper. In this letter we include a combination of previous comments from the reviewer as well as those from the current round. Text in italics are our responses to specific points. Underlined italicized text are our responses, and non-underlined italicized text are quotes from the paper.

Responses and Follow-up with the Reviewer From the Original Paper:

In the first round of feedback we received, we addressed the reviewer’s concerns based on the PLOS One criteria survey completed by the reviewer and extensively detailed the changes that we made in a detailed response letter. We also pointed to the specific changes we made in the manuscript, included with tracked changes. We have re-included our previous comments alongside the response to the current reviewer comments to demonstrate how they were addressed when necessary, as well as added new comments and questions for the reviewer based on the current reviews.

Journal Criteria: 1. Does the manuscript provide a valid rationale for the proposed study, with clearly identified and justified research questions?

Reviewer #1: Partly

Current Response: Our original response to this is below, and it is quite detailed. We feel that we have sufficiently addressed this point/rating. However, there is no specific information in the review that we could use to know what the reviewer wants. We have put a lot of effort into making this work clear based on the first round of feedback. If the reviewer wants additional, specific changes for clarification of this point, please let us know what those specific changes are. We feel the changes we have made to the manuscript make the research question and its justification very clear.

Previous Response: We thank the reviewer for this comment. As we edited the paper, we have made efforts to clarify our rationale for the proposed study. See pages 4 and pages 7-8 for two key examples of these changes:

(from page 4) “The prejudice and ideology literatures do not offer many concrete methodological suggestions for predicting the magnitude and direction of the effect of ideology on prejudice against specific groups. As an example, widely cited dual process models of prejudice suggest that a desire for social conformity and belief in a dangerous world predict right-wing authoritarian (RWA) attitudes, whereas the belief the world is a competitive jungle predicts social dominance, both of which predict prejudice [14]. Despite providing a theoretical rationale for the relationship between manifestations of ideology (i.e., RWA) and prejudice towards certain groups, this perspective does not provide scaffolding to predict the specific level of prejudice a person will exhibit towards a variety of target groups, in addition to examining a somewhat limited set of targets. This is a gap that should be filled because there is both practical and theoretical value in models that can predict the ideology-prejudice association across targets [4].”

(from pages 7-8) “The purpose of this project is to stress-test Brandt’s [4] original models using alternative measures of prejudice, including alternative explicit measures (i.e., relative measures of explicit prejudice), and reaction time measures (i.e., Implicit Association Test scores [20]). This addresses a key shortcoming of the original study. In particular, the original study used only feeling thermometers as an explicit measure of prejudice. Although feeling thermometers are common measures of prejudice (e.g., [10], [36], [18]), they are just one possible measure of group-based attitudes. The predictive accuracy of the models might be limited to feeling thermometers. If so, this would limit the utility of the predictive models to only studies that use feeling thermometers. This is a potentially substantial limitation given the diversity of prejudice measures that exist. However, if the predictive models work well with alternative measures of prejudice, i.e., the explicit and reaction time measures, it suggests that the model is much more generalizable. This would mean that this model could be used for different types of prejudice measures.”

Below we respond to criteria 2 and 3 simultaneously.

Journal Criteria: 2. Is the protocol technically sound and planned in a manner that will lead to a meaningful outcome and allow testing the stated hypotheses?

Reviewer #1: Partly

Journal Criteria: 3. Is the methodology feasible and described in sufficient detail to allow the work to be replicable?

Reviewer #1: No

Current Response: We made extensive changes to the manuscript based on the reviewer’s comments regarding our initial submission. These changes were documented with tracked changes in the re-submission. The edits we made included such changes as further explaining how missing data will be handled up to current standards, adding additional clarification about how we will compute MSEs, and highlighted the advantage of our methods. In addition, we provide the full code we will use to analyze the data on OSF, which is linked in the manuscript on page (https://osf.io/buwp7/?view_only=3905d5d1d54b4a499483c03a089b9f6e) so anybody can see exactly which analyses we will run and how. This aids with clarity and replicability. The Ideology 2.0 dataset is a professionally collected, gold-standard dataset. It seems highly unlikely results from these data will not be replicable. Below we also include a few examples of our previous response addressing issues regarding methodological clarification and replicability, the full responses to all points are in our initial rebuttal letter. Without additional specific information from the reviewer however, we do not know what the reviewer wants. If the reviewer would like additional, specific changes we request they clearly state exactly what they want. We feel we have already addressed all comments given by the reviewer at time 1 regarding the method and replicability, and that both are clear.

Examples Of Our Previous Response to This Comment: Additionally, we thank the reviewer for the request for clarification regarding post hoc tests and statistical software. We have added a clearer description of our strategy for our post hoc tests on page 21 (see quote below). For clarity, all analyses will be run using R Studio.

“Then, we will run a mixed ANOVA and post-hoc tests using the jmv package in R [30] to determine whether the differences between model MSEs are statistically significant (using an alpha level of .05). The model that has the lowest MSE (and the difference between that model and the next-lowest model is statistically significant) can be considered the most accurate predictive model of prejudice in this context. The calculations for the MSEs as well as the mixed ANOVA with post-hoc tests are included in the code on the project’s OSF page. The OSF Wiki contains details related to which lines of the code correspond to each calculation and test.”

Furthermore, we have added specific details about how MSEs are calculated on page 21 for the absolute measures (see quote below): 

“We will estimate the mean squared error (MSE) of the observed ideology-prejudice association compared to the predicted association. MSEs serve as a means of evaluating the accuracy of a predictive model by comparing predicted and actual values. To calculate the MSEs, we will take the actual ratings of prejudice across targets, subtract them from the estimates from the equations, square these values, and then take the average.”

The MSE calculations for the relative measures are described on page 31 (see quote below):

“After we have trained our models using the training data, we will use the test data (the remaining 20% of the Ideology 2.0 dataset, for each target group) to estimate the observed relationship between ideology and comparative prejudice for each target group pair and see which predictive model captures observed comparative prejudice most closely. We will do this by estimating the MSE of the observed ideology-comparative prejudice association compared to the predicted association. By comparing the MSEs of each of the five models predicting the participant ideology-prejudice relationship across targets and the observed values of prejudice against each target in the test set, we can test which model(s) are most effective in predicting prejudice across targets. Lower MSEs when comparing the slopes in the estimates to the observed levels of participant ideology-comparative prejudice indicate better fit in this context.”

Below are the new comments from the reviewer, from the second round of feedback. 

Reviewer #1: 

Point 1 - Methodolgy is tough to interpret. How can data be accessed as it is gathered in stages.

This is a registered report format https://genweb.plos.org/RR/EditorResources_ONERegisteredReports.pdf. In a registered report, the authors propose a study and method as well as detail the analysis to be conducted. After review, the paper may be accepted in-principle. This means that the journal agrees to publish the results, whatever they may be, if the authors undertake the proposed analyses in the agreed upon way. After in-principle acceptance the data are obtained, analyzed, and the manuscript is completed before being sent for review once more. In the paper, we describe on page 9 how the Ideology 2.0 data will be obtained. We are in contact with Schmidt and colleagues and they have agreed to provide us with the data upon stage 1 (in-principle) acceptance

“Here we propose testing four key models from Brandt [4], in addition to a null model (see Table 1) using a large dataset. Schmidt and colleagues [31] have issued a call for registered reports and have agreed to provide us with a new and very large (N = 261,119) dataset to conduct our analyses upon in-principle acceptance of a Stage 1 registered report. Studies 1 and 2 will use the Ideology 2.0 dataset [31].”

Furthermore, we describe how the data will be posted on OSF once it is obtained, and how it will be obtained on page 14.

“Across studies, we report how we determined sample size, how all participant exclusions were determined, and all manipulations and measures. All data used to derive this registered report and annotated R code (including proof-of-concept MSE estimates, which are described in the project OSF Wiki) are on the project’s OSF page: https://osf.io/buwp7/?view_only=3905d5d1d54b4a499483c03a089b9f6e. Upon acceptance of the Stage 1 registered report, we will pre-register the studies and planned analyses and then we will be given the full data. The full data will be posted on the project’s OSF page upon the final submission.”

Point 2 -Each data analytic step has not been meaningul explained e.g post-hoc tests were used by Turkey or Bonferroni?

We attempted to clarify this in the first revision by stating that we would use an alpha of .05 (not an adjusted p value). There are any number of procedures one might use when making post-hoc comparisons and Tukey and Bonferroni are but a few. We do not wish to use these strategies however, because we do not wish to omit potentially meaningful effects (i.e., we are more concerned about type 2 errors than type 1 errors). We have further clarified this on page 23 of the manuscript. 

“We will opt to use the original p values for the post-hoc tests as opposed to a procedure like Bonferroni or Tukey because we do not wish to omit potentially meaningful significant results based on stringent adjustments (Garcia-Perez, 2023).”

Point 3 -If data is not yet full or reached limit then must be waited for the complete findings.

Our response to this is above. We cannot obtain or analyze the full data yet, as this is a registered report. To do so would violate the guidelines of a registered report https://genweb.plos.org/RR/EditorResources_ONERegisteredReports.pdf. What we have done is detail all steps that will be used in the analysis of the data when we obtain it using a very small subset of the data made available to us for the purposes of writing this stage 1 registered report as pilot analyses are allowed in PLOS One’s guidelines. If the paper is accepted in principle, we can obtain the data and complete the work, but we cannot do so until we are at that stage.

---

## [Decision Letter · Decision Letter 2]

24 Jul 2024

Registered report protocol: Stress testing predictive models of ideological prejudice

PONE-D-23-11879R2

Dear Dr. Thompson,

We’re pleased to inform you that your manuscript has been judged scientifically suitable for publication and will be formally accepted for publication once it meets all outstanding technical requirements.

Kind regards,

Chetan Sinha

Academic Editor

PLOS ONE

Additional Editor Comments (optional):

Reviewers' comments:

Reviewer's Responses to Questions

**Comments to the Author**

1. Does the manuscript provide a valid rationale for the proposed study, with clearly identified and justified research questions?

Reviewer #1: Yes

2. Is the protocol technically sound and planned in a manner that will lead to a meaningful outcome and allow testing the stated hypotheses?

Reviewer #1: Yes

3. Is the methodology feasible and described in sufficient detail to allow the work to be replicable?

Reviewer #1: Yes

4. Have the authors described where all data underlying the findings will be made available when the study is complete?

Reviewer #1: Yes

5. Is the manuscript presented in an intelligible fashion and written in standard English?

Reviewer #1: Yes

6. Review Comments to the Author

You may also provide optional suggestions and comments to authors that they might find helpful in planning their study.

Reviewer #1: Dear authors,

Have gone through the major concern for the pilot data and is accessed in the OSF. It will be good to indicate completion of the prospective study that can be manifested in form of empirical article.

7. PLOS authors have the option to publish the peer review history of their article (what does this mean?). If published, this will include your full peer review and any attached files.

Reviewer #1: No

---

## [Editor Report · Acceptance letter]

20 Aug 2024

PONE-D-23-11879R2 

PLOS ONE

Dear Dr. Thompson, 

I'm pleased to inform you that your manuscript has been deemed suitable for publication in PLOS ONE. Congratulations! Your manuscript is now being handed over to our production team.

Kind regards, 

on behalf of

Dr. Chetan Sinha 

Academic Editor

PLOS ONE